# On Elimination Strategies for Bandit Fixed-Confidence Identification

**Andrea Tirinzoni**[*]
Meta AI
Paris, France
tirinzoni@fb.com

**Rémy Degenne**
Univ. Lille, Inria, CNRS, Centrale Lille, UMR 9189 CRIStAL, F-59000 Lille, France
remy.degenne@inria.fr

## Abstract

Elimination algorithms for bandit identification, which prune the plausible correct answers sequentially until only one remains, are computationally convenient since they reduce the problem size over time. However, existing elimination strategies are often not fully adaptive (they update their sampling rule infrequently) and are not easy to extend to combinatorial settings, where the set of answers is exponentially large in the problem dimension. On the other hand, most existing fully-adaptive strategies to tackle general identification problems are computationally demanding since they repeatedly test the correctness of every answer, without ever reducing the problem size. We show that adaptive methods can be modified to use elimination in both their stopping and sampling rules, hence obtaining the best of these two worlds: the algorithms (1) remain fully adaptive, (2) suffer a sample complexity that is never worse of their non-elimination counterpart, and (3) provably eliminate certain wrong answers early. We confirm these benefits experimentally, where elimination improves significantly the computational complexity of adaptive methods on common tasks like best-arm identification in linear bandits.

## 1 Introduction

The multi-armed bandit is a sequential decision-making task which is now extensively studied (see, e.g., [1] for a recent review). In this problem, an algorithm interacts with its environment by sequentially "pulling" one among $K \in \mathbb{N}$ arms and observing a sample from a corresponding distribution. Among the possible objectives, we focus on *fixed-confidence identification* [2, 3, 4, 5]. In this setting, the algorithm successively collects samples until it decides to stop and return an answer to a given query about the distributions. Its task is to return the correct answer with at most a given probability of error $\delta$, and its secondary goal is to do so while stopping as early as possible. This problem is called "fixed-confidence" as opposed to "fixed-budget", where the goal is to minimize the error probability with at most a given number of samples [6, 7, 8, 9, 10].

The most studied query is *best arm identification* (BAI), where the aim is to return the arm whose distribution has highest mean. A variant is Top-m identification [11], where the goal is to find the $m$ arms with highest means. While these are the most common, other queries have been studied, including thresholding bandits [9], minimum threshold [12], and multiple correct answers [13].

---

[*]Work done while at Inria Lille.

36th Conference on Neural Information Processing Systems (NeurIPS 2022).

Algorithms for fixed-confidence identification can be generally divided into two classes: those based on *adaptive sampling* and those based on *elimination*. Adaptive algorithms [e.g., 8, 11, 4, 14] update their sampling strategy at each round and typically stop when they can simultaneously assess the correctness of every answer. They often enjoy strong theoretical guarantees. For instance, some of them [4, 15, 16, 17] have been shown to be optimal as $\delta \to 0$. However, since they repeatedly test the correctness of every answer, they are often computationally demanding. Elimination-based strategies [e.g., 2, 18, 19, 20, 21] maintain a set of "active" answers (those that are still likely to be the correct one) and stop when only one remains. They typically update their sampling rules and/or the active answers infrequently. This, together with the fact that eliminations reduce the problem size over time, makes them more computationally efficient but also yields large sample complexity in practice. Moreover, while adaptive algorithms for general identification problems (i.e., with arbitrary queries) exist [4, 15, 17], elimination-based strategies are not easy to design at such a level of generality. In particular, they are not easy to extend to structured combinatorial problems (such as Top-m), where the number of answers is exponential in the problem dimension.[2]

In this paper, we design a novel elimination rule for general identification problems which we call *selective* elimination. It can be easily combined with existing adaptive strategies, both in their stopping and sampling rules, making them achieve the best properties of the two classes mentioned above. In particular, we prove that (1) selective elimination never suffers worse sample complexity than the original algorithm, and hence remain asymptotically optimal whenever the base algorithm is; (2) It provably discards some answers much earlier than the stopping time; (3) It improves the computational complexity of the original algorithm when some answers are eliminated early. Experimentally, we compare several existing algorithms for three identification problems (BAI, Top-m, and thresholding bandits) on two bandit structures (linear and unstructured). We find that, coherently across all experiments, existing adaptive strategies achieve significant gains in computation time and, to a smaller extent, in sample complexity when combined with selective elimination.

## 1.1 Bandit fixed-confidence identification

An algorithm interacts with an environment composed of $K > 1$ *arms*. At each time $t \in \mathbb{N}$, the algorithm picks an arm $k_t$ and observes $X_t^{k_t} \sim \nu_{k_t}$, where $\nu_{k_t}$ is the distribution of arm $k_t$. At a time $\tau$, the algorithm stops and returns an answer $\hat{\imath}$ from a finite set $\mathcal{I}$. Formally, let $\mathcal{F}_t$ be the $\sigma$-algebra generated by the observations up to time $t$. An identification algorithm is composed of

1. *Sampling rule*: the sequence $(k_t)_{t \in \mathbb{N}}$, where $k_t$ is $\mathcal{F}_{t-1}$-measurable.
2. *Stopping rule*: a stopping time $\tau$ with respect to $(\mathcal{F}_t)_{t \in \mathbb{N}}$ and a random variable $\hat{\imath} \in \mathcal{I}$, i.e., the answer returned when stopping at time $\tau$.

Note that, while it is common to decouple $\tau$ and $\hat{\imath}$, we group them to emphasize that the time at which an algorithm stops depends strongly on the answer it plans on returning.

We assume that the arm distributions depend on some unknown parameter $\theta \in \mathcal{M}$, where $\mathcal{M} \subseteq \mathbb{R}^d$ is the set of possible parameters, and write $\nu_k(\theta)$ for $k \in [K]$ to make this dependence explicit. For simplicity, we shall use $\theta$ to refer to the bandit problem $(\nu_k(\theta))_{k \in [K]}$. This assumption allows us to include linear bandits in our analysis. We let $i^\star : \mathcal{M} \to \mathcal{I}$ be the function, known to the algorithm, which returns the unique correct answer for each problem. The algorithm is correct on $\theta$ if $\hat{\imath} = i^\star(\theta)$.

**Definition 1.1** ($\delta$-correct algorithm)**.** *An algorithm is said to be $\delta$-correct on $\mathcal{M} \subseteq \mathbb{R}^d$ if for all $\theta \in \mathcal{M}$, $\tau < +\infty$ almost surely and $\mathbb{P}_\theta(\hat{\imath} \neq i^\star(\theta)) \leq \delta$ .*

We want to design algorithms that, given a value $\delta$, are $\delta$-correct on $\mathcal{M}$ and have minimal expected sample complexity $\mathbb{E}_\theta[\tau]$ for all $\theta \in \mathcal{M}$. A lower bound on $\mathbb{E}_\theta[\tau]$ was proved in [4]. In order to present it, we introduce the concept of *alternative* set to an answer $i \in \mathcal{I}$: $\Lambda(i) := \{\lambda \in \mathcal{M} \mid i^\star(\lambda) \neq i\}$, the set of parameters for which the correct answer is not $i$. Let us denote by $\mathrm{KL}_k(\theta, \lambda)$ the Kullback-Leibler (KL) divergence between the distribution of arm $k$ under $\theta$ and $\lambda$. Then the lower bound states that for any algorithm that is $\delta$-correct on $\mathcal{M}$ and any problem $\theta \in \mathcal{M}$,

$$\mathbb{E}_\theta[\tau] \geq \log(1/(2.4\delta))/H^\star(\theta) \text{ , with } H^\star(\theta) := \max_{\omega \in \Delta_K} \inf_{\lambda \in \Lambda(i^\star(\theta))} \sum_{k \in [K]} \omega^k \, \mathrm{KL}_k(\theta, \lambda) \text{ .} \quad (1)$$

---

[2]An elimination strategy for specific unstructured combinatorial problems has been introduced in [22].

**Example: BAI in Gaussian linear bandits**   While our results apply to general queries, we illustrate all statements of this paper on the widely-studied task of BAI in Gaussian linear bandits [19, 14, 23, 16, 24]. In this setting, each arm $k \in [K]$ has a Gaussian distribution $\mathcal{N}(\mu_k(\theta), 1)$ with mean $\mu_k(\theta) = \phi_k^\top \theta$, a linear function of the unknown parameter $\theta \in \mathbb{R}^d$ (and $\mathcal{M} = \mathbb{R}^d$) and of known arm features $\phi_k \in \mathbb{R}^d$. The set of answers is $\mathcal{I} = [K]$ and the correct answer is $i^\star(\theta) := \arg\max_{k \in [K]} \phi_k^\top \theta$.

Finally, for $x \in \mathbb{R}^d$ and $A \in \mathbb{R}^{d \times d}$, we define $\|x\|_A := \sqrt{x^\top A x}$. For $\omega \in \mathbb{R}^K$, let $V_\omega := \sum_{k=1}^K \omega^k \phi_k \phi_k^\top$. With this notation, we have $\sum_{k \in [K]} \omega^k \, \mathrm{KL}_k(\theta, \lambda) = \frac{1}{2} \|\theta - \lambda\|_{V_\omega}^2$.

## 1.2 Log-likelihood ratio stopping rules

Most existing adaptive algorithms use a log-likelihood ratio (LLR) test in order to decide when to stop. Informally, they check whether sufficient information has been collected to confidently discard at once all answers except one. Since such LLR tests are crucial for the design of our general elimination rules, we now describe their principle.

Given two parameters $\theta, \lambda \in \mathcal{M}$, the LLR of observations $X_{[t]} = (X_1^{k_1}, \dots, X_t^{k_t})$ between models $\theta$ and $\lambda$ is $L_t(\theta, \lambda) := \log \frac{d\mathbb{P}_\theta}{d\mathbb{P}_\lambda}(X_{[t]}) = \sum_{s=1}^t \log \frac{d\mathbb{P}_\theta}{d\mathbb{P}_\lambda}(X_s^{k_s})$. Let $\hat{\theta}_t := \arg\max_{\lambda \in \mathcal{M}} \log \mathbb{P}_\lambda(X_{[t]})$ be the maximum likelihood estimator of $\theta$ from $t$ observations. In Gaussian linear bandits, we have $L_t(\theta, \lambda) = \frac{1}{2} \|\theta - \lambda\|_{V_{N_t}}^2 + (\theta - \lambda)^\top V_{N_t}(\hat{\theta}_t - \theta)$, where $N_t^k := \sum_{s=1}^t \mathbb{1}(k_s = k)$. See Appendix C for more details. $L_t(\theta, \lambda)$ is closely related to $\sum_{k=1}^K N_t^k \, \mathrm{KL}_k(\theta, \lambda)$, a quantity that appears frequently in our results. Indeed, the difference between these quantities is a martingale, which is a lower order term compared to them. The LLR stopping rule was introduced to the bandit literature in [4]. At each step $t \in \mathbb{N}$, the algorithm computes the infimum LLR to the alternative set of $i^\star(\hat{\theta}_t)$ and stops if it exceeds a threshold, i.e., if

$$\inf_{\lambda \in \Lambda(i^\star(\hat{\theta}_t))} L_t(\hat{\theta}_t, \lambda) \geq \beta_{t,\delta} , \tag{2}$$

where the function $\beta_{t,\delta}$ can vary, notably based on the shape of the alternative sets. The recommendation rule is then $\hat{\imath} = i^\star(\hat{\theta}_t)$. Informally, the algorithm stops if it has enough information to exclude all points $\lambda$ for which the answer is not $i^\star(\hat{\theta}_t)$. This stopping rule enforces $\delta$-correctness, provided that the sampling rule ensures $\tau < +\infty$ a.s. and that $\beta_{t,\delta}$ is properly chosen. The most popular choice is to ensure a concentration property of $L_t(\hat{\theta}_t, \theta)$. For example, if for all $\delta$, $\beta_{t,\delta}$ guarantees that

$$\mathbb{P}\left(\exists t \geq 1 : L_t(\hat{\theta}_t, \theta) \geq \beta_{t,\delta}\right) \leq \delta, \tag{3}$$

LLR stopping with that threshold returns a wrong answer with probability at most $\delta$. Such concentration bounds can be found in [25, 26] for linear and unstructured bandits, respectively. This LLR stopping rule is used in many algorithms [4, 14, 15, 16, 24, 17][3]. Some of them have been proven to be *asymptotically optimal*: their sample complexity upper bound matches the lower bound (1) when $\delta \to 0$. However, improvements are still possible: their sample complexity for moderate $\delta$ may not be optimal and their computational complexity may be reduced, as we will see.

## 2 Elimination stopping rules for adaptive algorithms

We show how to modify the stopping rule of adaptive algorithms using LLR stopping to perform elimination. We assume that the alternatives sets $\Lambda(i)$ can be decomposed into a union of sets which we refer to as *alternative pieces* (or simply pieces), with the property that computing the infimum LLR over these sets is computationally easy.

**Assumption 2.1.** *For all $i \in \mathcal{I}$, there exist pieces $(\Lambda_p(i))_{p \in \mathcal{P}(i)}$, where $\mathcal{P}(i)$ is a finite set of piece indexes, such that $\Lambda(i) = \bigcup_{p \in \mathcal{P}(i)} \Lambda_p(i)$ and $\inf_{\lambda \in \Lambda_p(i)} L_t(\hat{\theta}_t, \lambda)$ can be efficiently computed for all $p \in \mathcal{P}(i)$ and $t > 0$.*

---

[3]LinGapE [14] does not use LLR stopping explicitly, but its stopping rule is equivalent to it. We can write it as: stop if for all points inside a confidence region a gap is small enough, that is if all those points do not belong to the alternative of $i^\star(\hat{\theta}_t)$. The contrapositive of that statement is exactly LLR stopping.

This assumption is satisfied in many problems of interest, including BAI, Top-$m$ identification, and thresholding bandits (see Appendix B). Indeed, in all applications we consider in this paper, the sets of Assumption 2.1 are half-spaces. In our linear BAI example, the piece indexes are simply arms. For $i, j \in [K]$ we can define $\Lambda_j(i) = \{\lambda \in \mathcal{M} \mid \phi_j^\top \lambda > \phi_i^\top \lambda\}$. Then, $\Lambda(i) = \bigcup_{j \in [K] \setminus \{i\}} \Lambda_j(i)$. Moreover, the infimum LLR (and the corresponding minimizer) can be computed in closed form as [e.g., 20] $\inf_{\lambda \in \Lambda_j(i)} L_t(\hat{\theta}_t, \lambda) = \max\{\hat{\theta}_t^T(\phi_i - \phi_j), 0\}^2 / \|\phi_i - \phi_j\|_{V_{N_t}^{-1}}^2$.

**Elimination stopping**   The main idea is that it is not necessary to exclude all $\Lambda_p(i)$ for $p \in \mathcal{P}(i)$ at the *same time*, as LLR stopping (2) does[4], in order to know that the algorithm can stop and return answer $i$. Instead, each piece can be pruned as soon as we have enough information to do so.

**Definition 2.2.** *A set $S \subseteq \mathbb{R}^d$ is said to be eliminated at time $t$ if, for all $\lambda \in S$, $L_t(\hat{\theta}_t, \lambda) \geq \beta_{t,\delta}$.*

From the concentration property (3), we obtain that the probability that $\theta \in S$ and $S$ is eliminated is less than $\delta$. LLR stopping interrupts the algorithm when the alternative set $\Lambda(i^\star(\hat{\theta}_t))$ can be eliminated. In elimination stopping, we eliminate smaller sets gradually, instead of the whole alternative at once. Formally, let us define, for all $i \in \mathcal{I}$,

$$\overline{\mathcal{P}}_t(i; \beta_{t,\delta}) = \left\{ p \in \mathcal{P}(i) : \inf_{\lambda \in \Lambda_p(i)} L_t(\hat{\theta}_t, \lambda) < \beta_{t,\delta} \right\} \tag{4}$$

as the subset of pieces for answer $i \in \mathcal{I}$ whose infimum LLR at time $t$ is below a threshold $\beta_{t,\delta}$. That is, the indexes of pieces that are *not* eliminated at time $t$. Moreover, we define, for all $i \in \mathcal{I}$, a set of *active pieces* $\mathcal{P}_t^{\text{stp}}(i)$ which is initialized as $\mathcal{P}_0^{\text{stp}}(i) = \mathcal{P}(i)$ (all piece indexes).

Our *selective elimination* rule updates, at each time $t$, only the active pieces of the empirical answer $i^\star(\hat{\theta}_t)$. That is, for $i = i^\star(\hat{\theta}_t)$, it sets

$$\mathcal{P}_t^{\text{stp}}(i) := \mathcal{P}_{t-1}^{\text{stp}}(i) \cap \overline{\mathcal{P}}_t(i; \beta_{t,\delta}), \tag{5}$$

while it sets $\mathcal{P}_t^{\text{stp}}(i) := \mathcal{P}_{t-1}^{\text{stp}}(i)$ for all $i \neq i^\star(\hat{\theta}_t)$. One might be wondering why not updating all answers at each round. The main reason is computational: as we better discuss at the end of this section, checking LLR stopping requires one minimization for *each* piece $p \in \mathcal{P}(i^\star(\hat{\theta}_t))$, while selective elimination requires only one for each *active* piece $p \in \mathcal{P}_{t-1}^{\text{stp}}(i^\star(\hat{\theta}_t))$. Thus, the latter becomes increasingly more computationally efficient as pieces are eliminated. For completeness, we also analyze the variant, that we call *full elimination*, which updates the active pieces according to (5) for *all* answers $i \in \mathcal{I}$ at each round. While we establish slightly better theoretical guarantees for this rule, it is computationally demanding and, as we shall see in our experiments, it does not significantly improve sample complexity w.r.t. selective elimination, which remains our recommended choice.

Let $\tau_{\text{s.elim}} = \inf_{t \geq 1} \{t \mid \mathcal{P}_t^{\text{stp}}(i^\star(\hat{\theta}_t)) = \emptyset\}$ and $\tau_{\text{f.elim}} := \inf_{t \geq 1} \{t \mid \exists i \in \mathcal{I} : \mathcal{P}_t^{\text{stp}}(i) = \emptyset\}$ be the stopping times of selective and full elimination, respectively. Intuitively, these two rules stop when one of the updated answers has all its pieces eliminated (and return that answer). We show that, as far as $\beta_{t,\delta}$ is chosen to ensure concentration of $\hat{\theta}_t$ to $\theta$, those two stopping rules are $\delta$-correct.

**Lemma 2.3** ($\delta$-correctness). *Suppose that $\beta_{t,\delta}$ guarantees (3) and that the algorithm verifies that, whenever it stops, there exists $i_\emptyset \in \mathcal{I}$ such that $\mathcal{P}_\tau^{\text{stp}}(i_\emptyset) = \emptyset$ and $\hat{i} = i_\emptyset$. Then, $\mathbb{P}_\theta(\hat{i} \neq i^\star(\theta)) \leq \delta$.*

All proofs for this section are in Appendix D. If an algorithm verifies the conditions of Lemma 2.3 and has a sampling rule that makes it stop almost surely, then it is $\delta$-correct. Interestingly, we can prove a stronger result than $\delta$-correctness: under the same sampling rule, the elimination stopping rules never trigger later than the LLR one *almost surely*. In other words, any algorithm equipped with elimination stopping suffers a sample complexity that is never worse than the one of the same algorithm equipped with LLR stopping. Let $\tau_{\text{llr}} := \inf_{t \geq 1} \{t \mid \inf_{\lambda \in \Lambda(i^\star(\hat{\theta}_t))} L_t(\hat{\theta}_t, \lambda) \geq \beta_{t,\delta}\}$.

**Theorem 2.4.** *For any sampling rule, almost surely $\tau_{\text{f.elim}} \leq \tau_{\text{s.elim}} \leq \tau_{\text{llr}}$.*

The proof of this theorem is very simple: if $\tau_{\text{llr}} = t$, then at $t$ all pieces $\Lambda_p(i^\star(\hat{\theta}_t))$ for $p \in \mathcal{P}(i^\star(\hat{\theta}_t))$ can be eliminated, hence $\tau_{\text{s.elim}} \leq t$. The proof that $\tau_{\text{f.elim}} \leq \tau_{\text{s.elim}}$ follows from the observation

---

[4]Under Assumption 2.1, LLR stopping is written as $\min_{p \in \mathcal{P}(i)} \inf_{\lambda \in \Lambda_p(i)} L_t(\hat{\theta}_t, \lambda) \geq \beta_{t,\delta}$ for $i = i^\star(\hat{\theta}_t)$, which implies that all alternative pieces of answer $i$ are discarded at once.

that full elimination always has less active pieces than selective elimination. Note that all three stopping rules must use the same threshold $\bar{\beta}_{t,\delta}$ to be comparable. Although simple, Theorem 2.4 has an important implication: we can take any existing algorithm that uses LLR stopping, equip it with elimination stopping instead, and obtain a new strategy that is never worse in terms of sample complexity and for which the original theoretical results on the stopping time still hold.

Finally, it is important to note that, while defining the elimination rule in the general form (5) allows us to unify many settings, storing/iterating over all sets $\mathcal{P}_t^{\text{stp}}(i)$ would be intractable in problems with large number of answers (e.g., top-m identification or thresholding bandits, where the latter is exponential in $K$). Fortunately, we show in Appendix B that this is not needed and efficient implementations exist for these problems that take only polynomial time and memory.

## 2.1 Elimination time of alternative pieces

We now show that elimination stopping can indeed discard certain alternative pieces much earlier that the stopping time. While all results so far hold for any distribution and bandit structure, in the remaining we focus on Gaussian linear bandits. Other distribution classes beyond Gaussians could be used with minor modifications (see Appendix C.2) but the Gaussian case simplifies the exposition. Since most existing adaptive sampling rules target the optimal proportions from the lower bound of [4], we unify them under the following assumption.

**Assumption 2.5.** *Consider the concentration events*

$$E_t := \left\{ \forall s \leq t : L_s(\hat{\theta}_s, \theta) \leq \beta_{t,1/t^2} \right\} . \tag{6}$$

*A sampling rule is said to have low information regret if there exists a problem-dependent function $R(\theta, t)$ which is sub-linear in $t$ such that for each time $t$ where $E_t$ holds,*

$$\inf_{\lambda \in \Lambda(i^\star(\theta))} \sum_{k \in [K]} N_t^k \, \mathrm{KL}_k(\theta, \lambda) \geq tH^\star(\theta) - R(\theta, t). \tag{7}$$

The left-hand side of (7) can be understood as the information collected by the sampling rule at time $t$ to discriminate $\theta$ with all its alternatives. Therefore, Assumption 2.5 requires that information to be comparable (up to a low-order term $R(\theta, t)$) with the maximal one from the lower bound. In Appendix F, we show that this is satisfied by both Track-and-Stop [4] and the approach in [15].

Let $H_p(\omega, \theta) := \inf_{\lambda \in \Lambda_p(i^\star(\theta))} \sum_{k \in [K]} \omega^k \, \mathrm{KL}_k(\theta, \lambda)$, the information that sampling with proportions $\omega$ brings to discriminate $\theta$ from the alternative piece $\Lambda_p(i^\star(\theta))$. Note that $H^\star(\theta) = \max_{\omega \in \Delta_K} \min_{p \in \mathcal{P}(i^\star(\theta))} H_p(\omega, \theta)$. For $\epsilon \geq 0$, let $\Omega_\epsilon(\theta) := \{\omega \in \Delta_K \mid \inf_{\lambda \in \Lambda(i^\star(\theta))} \sum_k \omega^k \, \mathrm{KL}_k(\theta, \lambda) \geq H^\star(\theta) - \epsilon\}$ be the set of $\epsilon$-optimal proportions.

**Theorem 2.6** (Piece elimination). *The stopping time of any sampling rule having low information regret, combined with LLR stopping, satisfies $\mathbb{E}[\tau] \leq \bar{t} + 2$, where $\bar{t}$ is the first integer such that*

$$t \geq \left( \left( \sqrt{\beta_{t,\delta}} + \sqrt{\beta_{t,1/t^2}} \right)^2 + R(\theta, t) \right) / H^\star(\theta). \tag{8}$$

*When the same sampling rule is combined with elimination stopping, let $\tau_p$ be the time at which $p \in \mathcal{P}(i^\star(\theta))$ is eliminated. Then, $\mathbb{E}[\tau_p] \leq \min\{\bar{t}_p, \bar{t}\} + 2$, where $\bar{t}_p$ is the first integer such that*

$$t \geq \max \left\{ \frac{\left( \sqrt{\beta_{t,\delta}} + \sqrt{\beta_{t,1/t^2}} \right)^2}{\min_{\omega \in \Omega_{R(\theta,t)/t}(\theta)} H_p(\omega, \theta)}, G(\theta, t) \right\}, \tag{9}$$

*with $G(\theta, t) = 0$ for full elimination and $G(\theta, t) = \frac{4\beta_{t,1/t^2} + R(\theta,t)}{H^\star(\theta)}$ for selective elimination.*

First, the bound we obtain on the elimination time of pieces in $\mathcal{P}(i^\star(\theta))$ is not worse than the bound we obtain on the stopping time of LLR stopping. Second, with elimination stopping, such eliminations can actually happen much sooner. Intuitively, sampling rules with low information regret play arms with proportions that are close to the optimal ones. If all of such "good" proportions provide large information for eliminating some piece $p \in \mathcal{P}(i^\star(\theta))$, then $p$ is eliminated much sooner than the actual stopping time (which requires eliminating the worst-case piece in the same set).

While both elimination rules are provably efficient, with full elimination enjoying slightly better guarantees[5], selective elimination provably never worsens (and possibly improves) the computational complexity over LLR stopping. In all applications we consider, implementing LLR stopping requires one minimization for each of the same alternative pieces we use for elimination stopping. Therefore, the total number of minimizations required by LLR stopping is $\sum_{t=1}^{\tau_{\mathrm{llr}}} |\mathcal{P}(i^\star(\hat{\theta}_t))|$ versus $\sum_{t=1}^{\tau_{\mathrm{s.elim}}} |\mathcal{P}_t^{\mathrm{stp}}(i^\star(\hat{\theta}_t))|$ for selective elimination. The second is never larger since $\tau_{\mathrm{s.elim}} \leq \tau_{\mathrm{llr}}$ by Theorem 2.4 and $\mathcal{P}_t^{\mathrm{stp}}(i^\star(\hat{\theta}_t)) \subseteq \mathcal{P}(i^\star(\hat{\theta}_t))$ for all $t$, and much smaller if eliminations happen early, as we shall verify in experiments. In our linear BAI example we need to perform $(K-1)$ minimizations at each step, one for each sub-optimal arm, in order to implement LLR stopping. On the other hand, we need only $|\mathcal{P}_t^{\mathrm{stp}}(i^\star(\hat{\theta}_t))|$ minimizations with selective elimination, one for each active sub-optimal arm, while full elimination takes $\sum_{i \in [K]} |\mathcal{P}_t^{\mathrm{stp}}(i)|$ to update all the sets.

Note that Theorem 2.6 does not provide a better bound on $\mathbb{E}[\tau]$ for elimination stopping than for LLR stopping. In fact, when evaluating the bound on $\mathbb{E}[\tau_p]$ for the worst-case piece in $p \in \mathcal{P}(i^\star(\theta))$, we recover the one on $\mathbb{E}[\tau]$. This is intuitive since the sampling rule is playing proportions that try to eliminate all alternative pieces at once. The following result formalizes this intuition.

**Theorem 2.7.** *Suppose that we can write $\beta_{t,\delta} = \log \frac{1}{\delta} + \xi(t,\delta)$ with $\lim_{\delta \to 0} \xi(t,\delta)/\log(1/\delta) = 0$. Then for any sampling rule that satisfies Assumption 2.5,*

$$\mathbb{E}[\tau_{\mathrm{llr}}] \leq \mathbb{E}[\tau_{\mathrm{elim}}] + f(\theta, \delta) .$$

*with $\lim_{\delta \to 0} f(\theta, \delta)/\log(1/\delta) = 0$. Here $\tau_{\mathrm{elim}}$ can stand for either full or selective elimination.*

See Appendix D.4 for $f$. This result shows that when the sampling rule is tailored to the LLR stopping rule, the expected LLR and elimination stopping times differ by at most low-order (in $\log(1/\delta)$) terms. As $\delta \to 0$ the two expected stopping times converge to the same value $H^\star(\theta)^{-1} \log(1/\delta)$, which is the asymptotically-optimal sample complexity prescribed by the lower bound (1).

We showed that, for both elimination rules, some pieces of the alternative are discarded sooner than the stopping time, and that the overall sample complexity of the method can only improve over LLR stopping. However, since the sampling rule of the algorithm was not changed, elimination does not change the computational cost of each sampling step, only the cost of checking the stopping rule.

## 2.2 An example

We compare LLR and elimination stopping on a simple example so as to better quantify the elimination times of Theorem 2.6 and their computational impact (see Appendix H.1 for a full discussion).

Consider BAI in a Gaussian linear bandit instance with unit variance, $d = 2$, and arbitrary number of arms $K \geq 3$ (see Figure 1). The arm features are $\phi_1 = (1,0)^T$, $\phi_2 = (0,1)^T$, and, for all $i = 3, \ldots, K$, $\phi_i = (a_i, b_i)^T$ with $a_i, b_i$ arbitrary values in $(-1, 0)$ such that $\|\phi_i\|_2 = 1$. The true parameter is $\theta = (1, 1-\varepsilon)^T$, for $\varepsilon \in (0, 1/2)$ a possibly very small value. Arm 1 is optimal with mean $\mu_1(\theta) = 1$, while arm 2 is sub-optimal with mean $\mu_2(\theta) = 1 - \varepsilon$. For all other arms $i = 3, \ldots, K$, $\mu_i(\theta) \leq 0$.

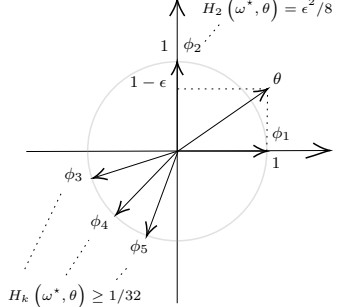

Let $\omega \in \Delta_K$ be any allocation. Recall that in BAI each piece index is simply an arm, and $\mathcal{P}(i^\star(\theta)) = \mathcal{P}(1) = \{2, \ldots, K\}$. Let $k \in \mathcal{P}(1)$ be any sub-optimal arm. The distance to the $k$-th alternative piece $H_k(\omega, \theta)$ can be computed in closed form as $H_k(\omega, \theta) = ((\phi_1 - \phi_k)^T \theta)^2/(2\|\phi_1 - \phi_k\|_{V_\omega^{-1}}^2)$. The optimal allocation is $\omega^\star = \arg\max_\omega \min_k H_k(\omega, \theta) = (1/2, 1/2, 0, \ldots, 0)^\top$.

Figure 1: Example of BAI instance with $d = 2$ and $K = 5$.

The intuition why this example is interesting is as follows. Any correct strategy is required to discriminate between arm 1 and 2 (i.e., to figure out that arm 1 is optimal), which requires roughly $O(1/\varepsilon^2)$ samples from both. An optimal strategy plays these two arms nearly with the same proportions. Since $\phi_1$ and $\phi_2$ form the canonical basis of $\mathbb{R}^2$, the samples collected by this strategy are informative for estimating the

---

[5]Note that $G(\theta, t)$ for selective elimination contributes only a finite (in $\delta$) sample complexity.

mean reward of *every* arm, even those than are not played. Then, since arms $3, \ldots, K$ have at least a sub-optimality gap of $1$, an elimination-based strategy discards them with a number of samples not scaling with $1/\varepsilon^2$. This means that a non-elimination strategy runs for $O(1/\varepsilon^2)$ steps over the original problem with $K$ arms, while an elimination-based one quickly reduces the problem to one with only $2$ arms. The main impact is computational: since most algorithms need to compute some statistics for each active arm at each round (e.g., closest alternatives, confidence intervals, etc.), the computational complexity of a non-elimination algorithm is at least $O(K/\varepsilon^2)$, while the one of an elimination-based variant is roughly $O(K + 1/\varepsilon^2)$, a potentially very large improvement.

We now quantify the elimination times and computational complexity on this example. Since such quantities depend on the specific sampling rule, we do it for an oracle strategy that samples according to $\omega^\star$. Similar results can be derived for any low information regret sampling rule (see Appendix H.1).

**Proposition 2.8.** *For any $K \geq 3$ and $\varepsilon \in (0, 1/2)$, for any $\delta \in (0, 1)$, the oracle strategy combined with LLR stopping satisfies on the example instance*

$$\mathbb{E}[\tau] \geq \Omega\left(\frac{\log(1/\delta)}{\varepsilon^2}\right).$$

*On the same instance, for the oracle strategy with elimination at stopping and a threshold $\beta_{t,\delta} = \log(1/\delta) + O(\log(t))$, the expected elimination time of any piece (i.e., arm) $k \geq 3$ is*

$$\mathbb{E}[\tau_k] \leq \widetilde{O}(\log(1/\delta)) \qquad \text{for full elimination,}$$

$$\mathbb{E}[\tau_k] \leq \widetilde{O}\left(\log(1/\delta) + \frac{1}{\varepsilon^2}\right) \qquad \text{for selective elimination.}$$

*Moreover, the expected per-round computation time of the oracle strategy with LLR stopping is $\Omega(K)$, while it is at most $O(K^2 \varepsilon^2)$ for full elimination and $O(K\varepsilon^2 + K/\log(1/\delta))$ for selective elimination.*

## 3 Elimination at sampling

We show how to adapt sampling rules in order to accommodate piece elimination. There are two reasons for doing this: first, adapting the sampling to ignore pieces that have been discarded could reduce the sample complexity; second, the amount of computations needed to update the sampling strategy is often proportional to the number of pieces and decreasing it can reduce the overall time.

We start from an algorithm using LLR stopping, for which we change the stopping rule as above. The sampling strategies that we can adapt are those that aggregate information from each alternative piece. For example, in linear BAI, methods that mimic the lower bound allocation (1), like Track-and-Stop [4], LinGame [16], or FWS [17], and even LinGapE [14], all compute distances or closest points to each piece in the decomposition $\{\lambda \mid \phi_j^\top \lambda \geq \phi_{i^\star(\hat{\theta}_t)}^\top \lambda\}$. Eliminating pieces at sampling simply means omitting from such computations the arms that were deemed sub-optimal. Algorithm 1 shows how Track-and-Stop [4] can be modified to incorporate elimination at sampling and stopping.

---

**Algorithm 1** Track-and-Stop [4]: vanilla (left) and with selective elimination (right)

| **while** not stopped **do** | **while** not stopped **do** |
|---|---|
| | Set $\mathcal{P}_t^{\text{stp}}(i^\star(\hat{\theta}_t)) = \mathcal{P}_{t-1}^{\text{stp}}(i^\star(\hat{\theta}_t))$ |
|     **for** $p \in \mathcal{P}(i^\star(\hat{\theta}_t))$ **do**     ▷ stopping |     **for** $p \in \mathcal{P}_{t-1}^{\text{stp}}(i^\star(\hat{\theta}_t))$ **do**     ▷ stopping |
|       $L_{p,t} = \inf_{\lambda \in \Lambda_p(i^\star(\hat{\theta}_t))} L_t(\hat{\theta}_t, \lambda)$ |       $L_{p,t} = \inf_{\lambda \in \Lambda_p(i^\star(\hat{\theta}_t))} L_t(\hat{\theta}_t, \lambda)$ |
| |       **if** $L_{p,t} > \beta_{t,\delta}$ **delete** $p$ from $\mathcal{P}_t^{\text{stp}}(i^\star(\hat{\theta}_t))$ |
|     **end for** |     **end for** |
|     **if** $\forall p \in \mathcal{P}(i^\star(\hat{\theta}_t)) : L_{p,t} > \beta_{t,\delta}$ **then** STOP |     **if** $\mathcal{P}_t^{\text{stp}}(i^\star(\hat{\theta}_t)) = \emptyset$ **then** STOP |
|     $w_t = \arg\max_\omega \min_{p \in \mathcal{P}(i^\star(\hat{\theta}_t))} H_p(\omega, \hat{\theta}_t)$ |     $w_t = \arg\max_\omega \min_{p \in \mathcal{P}_t^{\text{smp}}(i^\star(\hat{\theta}_t))} H_p(\omega, \hat{\theta}_t)$ |
|     **if** $\exists k : N_t^k < \sqrt{t}$ pull $k_{t+1} = \arg\min_k N_t^k$ |     **if** $\exists k : N_t^k < \sqrt{t}$ pull $k_{t+1} = \arg\min_k N_t^k$ |
|     **else** pull $k_{t+1} = \arg\min_k(N_t^k - tw_t^k)$ |     **else** pull $k_{t+1} = \arg\min_k(N_t^k - tw_t^k)$ |
| |     Update $\mathcal{P}_{t+1}^{\text{smp}}(i^\star(\hat{\theta}_t))$ (Algorithm 3) |
| **end while** | **end while** |

---

Similarly to elimination stopping, the idea is to maintain sets of active pieces at sampling $\mathcal{P}_t^{\text{smp}}(i)$ for each $i \in \mathcal{I}$. Note that these are different from the ones introduced in Section 2 for the stopping

rule. The set is updated at each step like we did for the stopping sets, but with a different threshold $\alpha_{t,\delta}$ (see Appendix E for details). Additionally, we reset it very infrequently at steps $t \in \{\bar{t}_0^{2^j}\}_{j \geq 0}$, where $\bar{t}_0 \geq 2$. Formally, let us define the helper sets $\tilde{\mathcal{P}}_t^{\mathrm{smp}}(i)$ as $\tilde{\mathcal{P}}_0^{\mathrm{smp}}(i) := \mathcal{P}(i)$ and

$$\tilde{\mathcal{P}}_t^{\mathrm{smp}}(i) := \begin{cases} \tilde{\mathcal{P}}_{t-1}^{\mathrm{smp}}(i) \cap \overline{\mathcal{P}}_t(i; \alpha_{t,\delta}) & \text{if } t \notin \{\bar{t}_0^{2^j}\}_{j \geq 0} \\ \overline{\mathcal{P}}_t(i; \alpha_{t,\delta}) & \text{otherwise}, \end{cases}$$

where $\overline{\mathcal{P}}_t$ was defined in (4). Let $\bar{t}_j := \bar{t}_0^{2^j}$ be the time step at which the $j$-th reset is performed and $j(t) := \lfloor \log_2 \log_{\bar{t}_0} t \rfloor$ be the index of the last reset before $t$. We define $\mathcal{P}_t^{\mathrm{smp}}(i) := \tilde{\mathcal{P}}_t^{\mathrm{smp}}(i) \cap \tilde{\mathcal{P}}_{\bar{t}_{j(t)}-1}^{\mathrm{smp}}(i)$, such that $\mathcal{P}_t^{\mathrm{smp}}(i)$ is the intersection of all active pieces from the second-last reset up to $t$, i.e., $\mathcal{P}_t^{\mathrm{smp}}(i) = \bigcap_{s=\bar{t}_{j(t)-1}}^t \overline{\mathcal{P}}_s(i; \alpha_{s,\delta})$. Since the resets are very infrequent, this definition only drops a small number of rounds from the intersection (less than $\sqrt{t}$). The detailed procedure to update these sets is summarized in Algorithm 3. As before, we can instantiate both selective and full elimination.

The reason for the resets is two-fold. First, they ensure that the algorithm stops almost surely as required by Definition 1.1. In fact, without resets, it might happen with some small (less than $\delta$) probability that pieces containing the true parameter are eliminated, in which case the sampling rule could diverge. Second, they guarantee that the thresholds $(\alpha_{s,\delta})_{s=\bar{t}_{j(t)-1}}^t$ used in $\mathcal{P}_t^{\mathrm{smp}}(i)$ are within a constant factor of each other. This is crucial to relate the LLR of active pieces at different times.

### 3.1 Properties

We consider a counterpart of Assumption 2.5 for sampling rules combined with piece elimination.

**Assumption 3.1.** *There exists a sub-linear (in $t$) problem-dependent function $R(\theta, t)$ such that, for each time $t$ where $E_t$ (defined in Equation 6) holds,*

$$\min_{p \in \mathcal{P}_t^{\mathrm{smp}}(i^\star(\theta))} H_p(N_t, \theta) \geq \max_{\omega \in \Delta_K} \sum_{s=1}^t \min_{p \in \mathcal{P}_{s-1}^{\mathrm{smp}}(i^\star(\theta))} H_p(\omega, \theta) - R(\theta, t).$$

Intuitively, the sampling rule maximizes the information for discriminating $\theta$ with all its alternatives from the sequence of active pieces $(\mathcal{P}_{s-1}^{\mathrm{smp}}(i^\star(\theta)))_{s=1}^t$. We prove in Appendix F that the algorithms for which we proved Assumption 2.5 also satisfy Assumption 3.1 when their sampling rules are combined with either full or selective elimination.

**Theorem 3.2.** *Consider a sampling rule that verifies Assumption 3.1 and uses either full or selective elimination with the sets $\mathcal{P}_t^{\mathrm{smp}}$. Then, Assumption 2.5 holds as well. Moreover, when using the same elimination rule at stopping, such a sampling rule verifies Theorem 2.6, i.e., it enjoys the same guarantees as without elimination at sampling.*

The proof is in Appendix E. Theorem 3.2 shows that for an algorithm using elimination at sampling and stopping, we get bounds on the times at which pieces of $\Lambda(i^\star(\theta))$ are discarded from the stopping rule which are not worse than those we obtained for the same algorithm without elimination at sampling. This result is non-trivial. We know that the sampling rule collects information to discriminate $\theta$ with its closest alternatives, and eliminating a piece cannot make the resulting "optimal" proportions worse at this task. However, it could make them worse at discriminating $\theta$ with alternatives that are not the closest. This would imply that the elimination times for certain pieces could actually increase w.r.t. not eliminating at sampling. Theorem 3.2 guarantees that this does not happen: eliminating pieces at sampling cannot worsen our guarantees. We shall see in our experiments that eliminating pieces in both the sampling and stopping rules often yields improved sample complexity.

## 4 Experiments

Our experiments aim at addressing the following questions: (1) how do existing adaptive strategies behave when combined with elimination at stopping and (when possible) at sampling? How do they compare with native elimination-based methods? (2) What is the difference between selective and full elimination? (3) How do LLR and elimination stopping compare as a function of $\delta$?[6]

---

[6]Our code is available at `https://github.com/AndreaTirinzoni/bandit-elimination`.

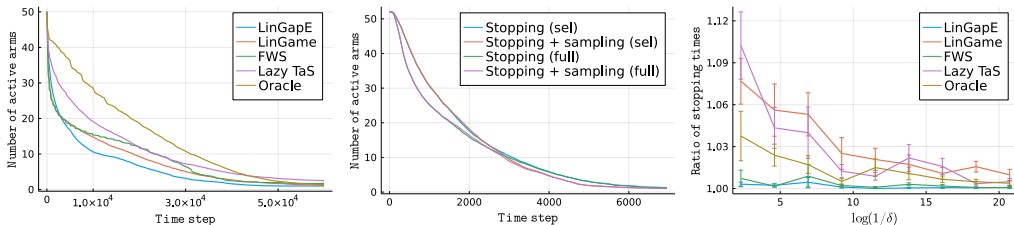

Figure 2: Experiments on linear instances with $K = 50$, $d = 10$, averaged over 100 runs, with the right plot showing standard deviations. (left) How different adaptive algorithms eliminate arms in BAI when using elimination stopping. (middle) LinGame on BAI when combined with full and selective elimination rules, either only at stopping or both at stopping and at sampling. (right) Ratio between the LLR and elimination stopping times of different algorithms as a function of $\log(1/\delta)$.

We ran experiments on two bandit structures: linear (where $d < K$) and unstructured (where $K = d$ and the arms are the canonical basis of $\mathbb{R}^d$). For each of them, we considered 3 pure exploration problems: BAI, Top-m, and online sign identification (OSI) [9, 27], also called thresholding bandits. All experiments use $\delta = 0.01$ and are averaged over 100 runs.

We combined adaptive algorithms which are natively based on LLR stopping with our elimination stopping rules and, whenever possible, we extended their sampling rule to use elimination. The selected baselines are the following. For linear BAI, LinGapE [14], LinGame [16], Frank-Wolfe Sampling (FWS) [17], Lazy Track-and-Stop (TaS) [24], XY-Adaptive [19], and RAGE [20] (the latter two are natively elimination based). For linear Top-m, m-LinGapE [28], MisLid [29], FWS, Lazy TaS[7], and LinGIFA [28]. For linear OSI, LinGapE[8], LinGame, FWS, and Lazy TaS. For unstructured instances linear algorithms are still applicable, and we further implemented LUCB [11], UGapE [8], and the Racing algorithm [18] for BAI and Top-m. We also tested an "oracle" sampling rule which uses the optimal proportions from the lower bound. Due to space constraints, we present only the results on linear structures. Those on unstructured problems can be found in Appendix G. The first experiments use randomly generated instances with $K = 50$ arms and dimension $d = 10$.

**Comparison of elimination times.** We analyze how different adaptive algorithms eliminate pieces when combined with selective elimination at stopping. To this purpose we focus on BAI, where the sets of pieces can be conveniently reduced to a set of active arms, those that are still likely to be the optimal one. Figure 2*(left)* shows how the set of active arms evolves over time for the 5 adaptive baselines. Notably, many arms are eliminated very quickly, with most baselines able to halve the set of active arms in the first 3000 steps. The problem size is quickly reduced over time. As we shall see in the last experiment, this will yield significant computational gains. We further note that the "oracle" strategy, which plays fixed proportions, seems the slowest at eliminating arms. The reason is that the optimal proportions from the lower bound focus on discriminating the "hardest" arms, while the extra randomization in adaptive rules might indeed eliminate certain "easier" arms sooner.

**Full versus selective elimination.** We combine the different algorithms with full and selective elimination, both at sampling and stopping. Due to space constrains, Figure 2*(middle)* shows the results only for LinGame (see Appendix G for the others). We note that full elimination seems faster at discarding arms in earlier steps, as we would expect theoretically. However, it never stops earlier than its selective counterpart. Moreover, its computational overhead is not advantageous. Overall, we concluded that our selective elimination rule is the best choice and we shall thus focus on it in the remaining. Finally, we remark that combining the sampling rule with elimination (no matter of what type) seems to discard arms faster in later steps, and could eventually make the algorithm stop sooner.

**LLR versus elimination stopping.** We now compare LLR and elimination stopping as a function of $\delta$. We know from theory that both stopping rules allow to achieve asymptotic optimality. Hence for asymptotically optimal sampling rules the resulting stopping times with LLR and elimination should tend to the same quantity as $\delta \to 0$. Figure 2*(right)*, where we report the ratio between the LLR stopping time and the elimination one for different algorithms, confirms that this is the case. Some algorithms (LinGapE and FWS) seem to benefit less from elimination stopping than the others,

---

[7]Lazy TaS, while analyzed only for BAI, can be applied to any problem since it is a variant of Track-and-Stop.
[8]LinGapE was originally proposed only for BAI in [14], but its extension to OSI is trivial.

| | Algorithm | No elimination (LLR) | | Elim. stopping | | Elim. stopping + sampling | |
|---|---|---|---|---|---|---|---|
| | | Samples | Time | Samples | Time | Samples | Time |
| BAI | LinGapE | $33.19 \pm 8.7$ | 0.23 | $33.11 \pm 8.7$ | 0.2 | $29.89 \pm 8.6$ | $0.18(-22\%)$ |
| | LinGame | $45.34 \pm 14.2$ | 0.23 | $43.67 \pm 13.4$ | 0.21 | $32.49 \pm 8.1$ | $0.18(-22\%)$ |
| | FWS | $42.26 \pm 60.1$ | 0.73 | $42.25 \pm 60.1$ | 0.7 | $32.62 \pm 18.0$ | $0.45(-38\%)$ |
| | Lazy TaS | $76.33 \pm 65.8$ | 0.15 | $74.08 \pm 65.8$ | 0.13 | $64.48 \pm 81.8$ | $0.12(-20\%)$ |
| | Oracle | $56.36 \pm 9.1$ | 0.05 | $55.36 \pm 9.3$ | 0.02 | | |
| | XY-Adaptive | | | | | $87.08 \pm 29.1$ | 0.44 |
| | RAGE | | | | | $106.87 \pm 30.7$ | 0.02 |
| Top-m ($m = 5$) | m-LinGapE | $63.69 \pm 11.1$ | 0.56 | $63.48 \pm 11.0$ | 0.41 | $59.57 \pm 9.4$ | $0.24(-57\%)$ |
| | MisLid | $87.77 \pm 20.4$ | 0.55 | $85.95 \pm 20.5$ | 0.4 | $69.58 \pm 16.0$ | $0.25(-55\%)$ |
| | FWS | $78.28 \pm 65.0$ | 3.0 | $78.23 \pm 65.0$ | 2.85 | $77.79 \pm 65.0$ | $0.97(-67\%)$ |
| | Lazy TaS | $161.43 \pm 96.9$ | 0.57 | $159.86 \pm 96.9$ | 0.43 | $146.06 \pm 82.6$ | $0.36(-36\%)$ |
| | Oracle | $102.45 \pm 16.1$ | 0.2 | $101.53 \pm 16.4$ | 0.08 | | |
| | LinGIFA | $58.31 \pm 10.8$ | 2.46 | $58.31 \pm 10.8$ | 2.33 | | |
| OSI | LinGapE | $17.31 \pm 2.3$ | 0.22 | $17.29 \pm 2.2$ | 0.19 | $14.71 \pm 2.0$ | $0.17(-23\%)$ |
| | LinGame | $23.77 \pm 4.1$ | 0.25 | $23.05 \pm 3.9$ | 0.21 | $14.87 \pm 2.0$ | $0.19(-24\%)$ |
| | FWS | $15.26 \pm 2.0$ | 0.83 | $15.24 \pm 2.0$ | 0.81 | $14.99 \pm 2.1$ | $0.56(-32\%)$ |
| | Lazy TaS | $35.11 \pm 10.2$ | 0.32 | $33.98 \pm 9.7$ | 0.3 | $23.51 \pm 5.6$ | $0.24(-25\%)$ |
| | Oracle | $29.1 \pm 4.8$ | 0.06 | $28.65 \pm 5.0$ | 0.03 | | |

Table 1: Experiments on linear instances with $K = 50$, $d = 20$. The "Time" columns report average times per iteration in milliseconds. The percentage in the last column is the change w.r.t. the time without elimination. Each entry reports the mean across 100 runs plus/minus standard deviation (which is omitted for compute times due to space constraints). Algorithms for which the third column is missing cannot be combined with elimination at sampling, while algorithms for which the first two columns are missing are natively elimination-based. Samples are scaled down by a factor $10^3$.

i.e., they achieve smaller ratios of stopping times. We believe this to be a consequence of their mostly "greedy" nature, while the extra randomization of the other algorithms might help in this aspect.

**Sample complexities and computation times.** We finally compare our baselines in all three exploration tasks, in terms of sample complexity and computation time. For this experiment, we selected a larger linear instance with $K = 50$ and $d = 20$, randomly generated (see the protocol in Appendix G). From the results in Table 1, we highlight three points. (1) The computation times of all adaptive algorithms decrease when using selective elimination stopping instead of LLR and further decrease when also using elimination at sampling. In the case of Top-m (i.e., the hardest combinatorial problem), most adaptive algorithms become at least twice faster with elimination at stopping and sampling instead of LLR. (2) Elimination at sampling improves the sample complexity of all algorithms. (3) For BAI, the natively elimination-based algorithm RAGE, which updates its strategy infrequently, is the fastest in terms of computation time but the slowest in terms of samples. Adaptive algorithms using elimination achieve run times that are within an order of magnitude of those of RAGE, while outperforming it in terms of sample complexity by a factor 2 to 3.

## 5 Conclusion

We proposed a selective elimination rule, which successively prunes the pieces of the empirical answer, that can be easily combined with existing adaptive algorithms for general identification problems. We proved that it reduces their computational complexity, it never worsens their sample complexity guarantees, and it provably discards certain answers early. Our experiments on different pure exploration problems and bandit structures show that existing adaptive algorithms often benefit from a reduced sample complexity when combined with selective elimination, while achieving significant gains in computation time. Moreover, they show that selective elimination is overall better (in terms of samples vs time) than its full variant which repeatedly updates the pieces of all answers.

Interesting directions for future work include investigating whether better guarantees on the stopping time can be derived for algorithms combined with elimination as compared to their LLR counterparts, and designing adaptive algorithms which are specifically tailored for elimination.

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
