# OpenReview forum: "On Elimination Strategies for Bandit Fixed-Confidence Identification"
_NeurIPS.cc/2022/Conference — NeurIPS 2022 Accept_

### Official Review · Reviewer_rbgz · 2022-07-01

**Rating:** 6
**Confidence:** 4
**Soundness:** 3 good
**Presentation:** 3 good
**Contribution:** 3 good

**Summary:**

The paper considers the problem of finding (or identifying) the correct answer to a specific question in a bandit scenario. Examples are what is the arm with the highest mean (best arm identification), or which arms have a mean larger than some specific value (thresholding bandits)? For this purpose, they revisit the algorithmic approaches based on log-likelihood ratio stopping, which are theoretically quite appealing but are oftentimes computationally expensive. Roughly speaking, these approaches stop as soon as there is an answer such that the alternatives to these answers are not plausible enough (measured in terms of minimal distance of log-likelihood ratio). Under the assumption that the underlying bandit problem scenario allows a decomposition of the possible alternative sets into smaller subsets (called pieces), the authors suggest two stopping criteria that have provably a smaller stopping time than the log-likelihood approach provided all are using the same sampling strategy. These stopping criteria are combining the log-likelihood ratio stopping approach with the non-adaptive elimination-based approaches by excluding pieces as soon as these are ``redundant’’ for the decision making. The authors show that for a couple of Gaussian linear bandit problems the assumption above is satisfied and the minimizer of the log-likelihood ratio can be efficiently computed on the corresponding pieces. Moreover, a bound in expectation on the resulting stopping times are shown, when the stopping criteria are combined with an efficient sampling rule.
Inspired by the suggested stopping criteria the authors derive a sampling strategy that is specifically tailored towards an early stopping of the former. The suggested approaches are investigated in an experimental study on synthetic data.

**Questions:**

- Are there bandit settings where it is not possible to decompose the alternative set into meaningful pieces? If yes, these should be mentioned in the paper.

- Of course, one might ask why the full elimination approach is considered at all, if it has such a high runtime, while the sampling complexity in the experiments is not significantly different compared to the LLR approach. In this way, the authors could save some space, which could be used to discuss other relevant things, e.g., describe how to apply the methods to other bandit settings than Gaussian linear bandits.





**Limitations:**

I would have appreciated if the authors would give a remark regarding a direct comparison of the actual stopping times, i.e. in which case it is possible and in which it is not. I think only for Theorem 2.6 it is.



**Strengths And Weaknesses:**

# Strengths

## Quite interesting topic
The underlying problem setting of general identification in bandits is a relevant theoretical problem scenario covering a range of practical applications and has already been investigated by a couple of several authors. The underlying research question of whether the fully adaptive approaches can be adapted such that they are computationally more efficient such as the non-adaptive elimination strategies is of utmost relevance for practical applications. The key challenge is of course to maintain the appealing theoretical properties of the fully adaptive approaches.

## Soundness
Overall, most of the results are sound as well as their proofs. I haven’t checked all proofs in every last detail but had a thorough look at the proofs of core results, which were fine as far as I can tell.

## Quality of writing/presentation
In general, the paper is well written. Although it is a rather technical paper the notation is well thought out.

# Weaknesses

## Modest theoretical contribution
Although the paper contains a fairly extensive appendix of proofs of the theoretical results, these are largely based on prior work and no groundbreaking new ideas come in, as far as I can tell. Therefore, I think the theoretical contribution of this paper is okay, but not significant.


## Vague description of eliminating at sampling
The part about the suggested sampling strategy can be improved regarding its clarity. To be more precise, I would have appreciated if the eliminating at sampling approach would have been presented in form of a pseudo-code instead of a rather vague verbal description.

---

> ### Author Response · Authors · 2022-08-01
> **Answer to Reviewer rbgz**
>
> We thank you for your time reviewing our paper and for your suggestions. There were two points raised by most reviewers, which we answered in a general comment: please read it first, then read this answer.
>
> **About being able to decompose the alternative set into meaningful pieces**
>
> This is true in any best arm identification task, top-m identification, ranking or thresholding bandits. We are not aware of an identification task in which the alternative set cannot be written as a union over arms or pairs of arms. Morally, for the LLR to be computable, the solution of its minimization over some elementary piece of the alternative has to be easily computable. We then take those pieces as our decomposition.
>
> **About going beyond gaussian bandits**
>
> We have a subsection about that question in appendix C.2. In a nutshell, the manipulations we do using likelihood ratios and KL divergences can be extended to other one-parameter exponential families, as was done in the paper “Non-asymptotic pure exploration by solving games”, by Degenne et al., under some restrictions on the family or the range of the parameters.
>
> **Direct comparison of stopping times**
>
> While we can compare the stopping times directly in Theorem 2.4 (instead of bounds on their expectations), being able to compare the performance of two algorithms in this very strong “almost sure” manner is highly unusual, and that is the only place in the paper where we are able to do so. When we introduce elimination at sampling, we are deeply changing the way the algorithm behaves. We are still able to obtain a comparison between the new version and the baseline based on bounds on their expected sample complexities, which is the standard way to compare algorithms across the bandit literature.
>
> **About our theoretical contributions**
>
> We agree that the techniques we used to prove some results about LinGame or TaS are not new, but we are after all adapting existing algorithms, and the reason the modified versions verify sample complexity bounds are close to the reasons behind the bounds of the original algorithms. However, the formulation of Assumption 3.1 (about elimination at sampling) and the fact that it can give sample complexity bounds for algorithms which use a constantly changing alternative set are completely new.
>
> Still about theoretical novelties, we want to highlight that the idea that we could improve upon LLR stopping is also innovative: most papers about BAI algorithms (all the ones about TaS, LinGame, FWS, and their derivatives in other settings) dismiss the question of the stopping rule by stating that LLR stopping allows asymptotically optimal algorithms. Of course, asymptotically LLR stopping is optimal and there is nothing better to do in the limit $\delta\to 0$, but we show that another choice also allows optimal algorithms, while being empirically better for both sample complexity when $\delta$ is large (only slightly better, around 10% in Table 1 for $\delta=0.01$) and computational complexity (significantly better, up to 50% improvement for our top-m experiment). We reopen the question of what a good stopping rule should do.
>
> We hope this answers some of your questions and we thank you again for your review.

---

> > ### Comment · Reviewer_rbgz · 2022-08-08
> > **RE: Answer to Reviewer rbgz**
> >
> > Dear authors,
> >
> > thank you very much for the detailed answer. However, I will still keep my score as it is, as I think the overall theoretical contribution is modest.

---

### Official Review · Reviewer_FuX3 · 2022-07-11

**Rating:** 5
**Confidence:** 4
**Soundness:** 3 good
**Presentation:** 2 fair
**Contribution:** 3 good

**Summary:**

 The authors propose a new algorithm design principle for elimination algorithms in the fixed-confidence, best-arm-identification setting with parametric distributions that modifies the usual stopping rule of track-and-stop-style algorithms (adaptive in their terminology).

Roughly, under an assumption that the parameter space of each alternative for each arm i (i.e. the parameters that would result in any arm other than i being the best) can be decomposed into a small subset of pieces that are easy to optimize over, the authors propose elimination of individual pieces. In contrast, the typical track-and-stop algorithm will continue until all pieces except for those belonging to a single action can be eliminated. This procedure can reduce the cost of checking the stopping rule, and is shown to have the same guarantees (correctness and asymptotic efficiency).

The authors then present sampling rule that also takes into account pieces that have been eliminated and show that, under a technical assumption, the new sampling rule does not degrade the sample complexity. Improved performance in both sample complexity and time complexity is shows experimentally.



**Questions:**

Can you provide some justification for your assumptions?Are there special cases when you can provably show an improved sample complexity?

**Limitations:**

I think a closer discussion about the assumptions is needed. Societal impact was not addressed, but I don't think it's necessary.

**Strengths And Weaknesses:**

While the writing was generally high quality, I found the overall presentation confusing and the theoretical results a bit underwhelming (though I didn't read the appendix, so it is possible the technical difficulties were larger than I thought).

First, I found the main ideas difficult to understand and had to work through section 2 a few times. I think the presentation could be improved if the section was restructured to highlight a cartoon of the original track-and-stop and introduce your new ideas an modifications. It wasn't immediately clear to me if your procedure would produce an action elimination algorithm or an adaptive algorithm. Beginning with the explanation that you are in the parametric setting and are modifying track and stop would help.

I also wish that the elimination sampling algorithm was presented more clearly, perhaps with pseudo-code (at least for your favorite instantiation of an adaptive algorithm).

It might be good to give a bit more intuition about what "adaptive sampling" and "elimination" algorithms do and their differences, especially since there are several well known algorithms (e.g. LUCB, TS variant)) that don't fit into your two classes. Specifically, I found your "repeatedly test the correctness of every answer" confusing, and it would be probably clearer to describe how elimination algorithms work in phases/epochs. I might go so far at to argue that "adaptive" is a confusing choice, as elimination algorithms also change their sampling distributions as data are collected (albeit usually through eliminating actions). Perhaps phased / non-phased would be a clearer distinction.

Another very import distinction is that "elimination algorithms" are generally non-parametric, needing only some way to get finite-sample confidence intervals on the target parameters, whereas "adaptive" methods require strong parametric assumptions.

I wish there was theory showing improvement in the sample complexity, and I wish there was a discussion about the assumptions (specifically, an argument about why they should be necessary).

Other feedback:
line 137: bad grammar

---

> ### Author Response · Authors · 2022-08-01
> **Answer to Reviewer FuX3**
>
> We thank you for your time reviewing our paper and for your suggestions. There were two points raised by most reviewers, which we answered in a general comment: please read it first, then read this answer.
>
> **Quality of the presentation of elimination**
>
> This was a common remark of all reviewers, and we changed the way we present the section about elimination at sampling (see the general comment to all reviewers for more details). In particular, we included a pseudo-code showing how to adapt track-and-stop as you suggested, and we thank you for the suggestion. A code for the adaptation of LinGame is also available in appendix I.
>
> **About adaptive vs elimination algorithms and the names of these categories**
>
> Indeed we agree that phased/non-phased could have been another way to describe the two families. We used “elimination” to mean phased algorithms in which the sampling proportions change only between phases, through the elimination process. Other algorithms, which constantly change the way they sample, we call “adaptive”. You mentioned LUCB: it is an adaptive algorithm in our classification and since it uses LLR stopping (for the same reason as what is explained in the footnote on page 3 about LinGapE), we can adapt it to use elimination at stopping or sampling. You can find experiments using modified LUCB in appendix G.3.2. The modified algorithms we create are still in the “adaptive” category.
>
> **About the assumptions**
>
> - Assumption 2.1 states that we can decompose the alternative into a union of pieces. This is true in any best arm identification task, top-m identification, ranking or thresholding bandits. We are not aware of an identification task in which the alternative set cannot be written as a union over arms or pairs of arms.
>
> - Assumptions 2.5 and 3.1 (that 3.1 number refers to the revised paper): we wanted to derive guarantees for modified versions of adaptive algorithms, but that is only possible if the algorithm we start from is “good” in some sense and we cannot prove anything on the modified algorithm without assumptions on the baseline. We extracted properties verified by some existing algorithms (as proved in appendix F) and made them into those assumptions. Let’s look at assumption 2.5 more precisely. The LLR stopping rule compares a random quantity to a threshold. Assumption 2.5 states roughly that the algorithm samples in a way that makes that quantity as large as possible.
>
> - The assumption on the threshold, which is now in appendix E, is discussed in appendix E.2. We needed to know how the threshold varies with t to calibrate the resets of the sampling sets and we made the assumption that it was $O(\log t)$, as in commonly used thresholds for linear bandits, but we could adapt that calibration to other increases like $O(\log\log t)$ without issue.
>
> **About showing improvements in sample complexity**
>
> Unfortunately, we believe that this is extremely complicated, at least with current techniques. The issue here is that some non-modified algorithms like track-and-stop are asymptotically optimal. Any improvement in sample complexity we could prove would necessarily disappear as delta goes to 0, hence we would need to provide a finite-delta analysis. However, the analysis of track-and-stop itself is mostly asymptotic, as are the existing proof techniques for LinGame, FWS or similar methods (the bound for LinGame is strictly speaking non-asymptotic but has very large terms that don’t reflect the performance of the algorithm, and the bound is tight only as $\delta$ goes to 0). In summary, the analysis of the baseline methods is not advanced enough to catch non-asymptotic effects, and any improvement we could show would have to be a non-asymptotic effect on low order (in $\log(1/\delta)$) terms.
>
> We want to insist on the fact that we still provide interesting theoretical guarantees: the fact that we can change the sampling rule to incorporate elimination while getting the same sample complexity bound is not trivial, since it modifies deeply the way the algorithm behaves. This property allows us to say that there is no risk for the sample complexity in using elimination at sampling, while on the other hand we can prove gains in computational complexity. See Appendix H in our revised manuscript for a specific instance where we can concretely quantify the improvement in computational complexity for a specific algorithm (Proposition H.4). And empirically, the gain of computational complexity is significant, around 50% on our top-m experiments (see table 1, page 9).
>
> We hope this answers some of your questions and we thank you again for your review.

---

> > ### Comment · Reviewer_FuX3 · 2022-08-07
> > **response**
> >
> > Thanks for the response. I think the new algorithm makes the presentation much clearer. I also find the author's argument that showing a general improvement in the sample complexity to be more difficult that I thought. Having an example where the improvement can be quantified is good enough for me. I've increased my score.

---

### Official Review · Reviewer_gFGn · 2022-07-12

**Rating:** 5
**Confidence:** 3
**Soundness:** 3 good
**Presentation:** 2 fair
**Contribution:** 3 good

**Summary:**

This work bridges the gap between elimination algorithms and fully adaptive algorithms for identification problems in bandits. One the one hand, the main strength and weakness of elimination algorithms are their high computational efficiency and large sample complexities, respectively. One the other hand, fully adaptive algorithms enjoy stronger sample complexity guarantees while being computationally inefficient. In this paper are designed fully adaptive algorithms that perform elimination. These algorithms are shown to get the best of both worlds, both theoretically and empirically.


**Questions:**

- In practice how does one modify an existing algorithm (say e.g. Track-and-Stop) to incorporate any of the elimination steps suggested in this work?
- Writing the infimum LLR in closed form is very illustrative. Are there settings where the quantities in the theorems are closed form? If so, would it be possible to add some more explicit statements?
- Are there original algorithms and new elimination counterparts of these for which the computational complexity can be computed? The claimed improvement on computational complexity feels very empirical, instead of being theoretically characterized.


**Limitations:**

I did not really find a clear statement about the limitations of the work. The authors state that the potential negative societal impacts of their work is N/A. This work is mainly theoretical, but it might still be valuable to mention how bandit algorithms are applied to the world and e.g. what could go wrong if the suggested algorithms are used.


**Strengths And Weaknesses:**

Strengths:
- The framework has a broad range of applications: it covers among others the problems of gaussian bandits, top-k, thresholding identification for linear and unstructured bandits.
- On this general setup, this work successfully tackles the problem of designing strategies that are both fully adaptive and performing elimination.
- The experiments support the theoretical claims.

Weaknesses:
- This paper could benefit from a better presentation:
- It would be nice to highlight a pseudo-code of the algorithm designed.
- It takes a long time to get to the main theoretical results.
- The theorems are hard to parse without examples/instantiations.

---

> ### Author Response · Authors · 2022-08-01
> **Answer to Reviewer gFGn**
>
> We thank you for your time reviewing our paper and for your suggestions. There were two points raised by most reviewers, which we answered in a general comment: please read it first, then read this answer.
>
> > In practice how does one modify an existing algorithm (say e.g. Track-and-Stop) to incorporate any of the elimination steps suggested in this work?
>
> In Section 3.1 of the revised manuscript, we added the pseudo-code of Track-and-Stop with and without elimination (Algorithm 1), which is intended as a general example of how existing algorithms can be easily modified. In summary, all the algorithms we consider compute the infimum LLR to the alternative set and sample arms so as to make it large (eg, by tracking the optimal allocation from the lower bound). The infimum LLR is in turn computed as a minimum over some pieces (eg, half-spaces associated to each arm in BAI). Modifying such algorithms to use eliminations is then very easy: it is enough to exclude the eliminated pieces from the computation of this minimum. We also refer the reviewer to Algorithm 2 in Appendix I, where we show how LinGame can be modified with elimination.
>
> > Are there settings where the quantities in the theorems are closed form?
>
> Unfortunately, quantities like the optimal value $H^\star(\theta)$ from the lower bound do not admit a closed-form expression in general, not even in “simple” problems like linear BAI. However, to provide more explicit statements, we added in Appendix H a specific instance of a linear BAI problem (described in Appendix H.1) where the quantities appearing in the bounds of Theorem 2.6 can indeed be computed. In particular, for this instance, we prove that the elimination time of most arms is significantly smaller than the actual stopping time (see Proposition H.5). See also the answer to Reviewer W482 for more details on this point.
>
> > Are there original algorithms and new elimination counterparts of these for which the computational complexity can be computed?
>
> To quantify concretely how much the computational complexity of an algorithm can be reduced with elimination, in Appendix H.2 we analyzed an oracle strategy which plays the optimal proportions from the lower bound in the specific instance of Appendix H.1. We chose this strategy because the sampling rule does not involve any complicated computation, and the computational complexity is dominated by the one of the stopping rule. For this strategy, we quantify in Proposition H.4 the reduction in compute time that elimination stopping brings over LLR, essentially showing that the former can be made arbitrarily faster as the problem becomes more complex (by decreasing the parameter $\epsilon$).

---

### Official Review · Reviewer_W482 · 2022-07-13

**Rating:** 6
**Confidence:** 4
**Soundness:** 3 good
**Presentation:** 2 fair
**Contribution:** 2 fair

**Summary:**

The paper proposes elimination rules that can be easily combined with existing fixed-confidence algorithms for pure exploration, both for sampling and stopping strategies. The benefits are improved performance in the nonasymptotic regime. The authors show a result that the elimination stopping results in eliminating pieces no later than the standard LLR stopping.

**Questions:**

Theorem 2.6. I understand that with elimination stopping it is never worse, but it is not clear if it makes a meaningful difference. I was not able to pin down on exactly how (9) can be strictly better than (8). I don't have a good sense on whether $H^*(\theta)$ is related to $\min_{...} H_p(\omega,\theta)$ in (9). I'd like to hear from the authors on this. Without a careful comparison, I cannot say for sure that it is meaningful. To make a stronger argument, I would find a specific instance where (8) can be orderwise larger than (9) to showcase the real difference.

The explanation of reset times in Section 3.1 was not clear at all to me. What is the role of the resetting? It would  be beneficial if we have a protocol/pseudocode description of what we expect from existing algorithms and what we do for eliminating the pieces, side by side. Also, please provide a specific example algorithm where you show how elimination pieces at sampling can be combined.

How existing elimination strategies are not easy to extend to combinatorial settings: I think Successive Accept and Reject (SAR) was applied to combinatorial pure exploration in Chen et al., Combinatorial Pure Exploration of Multi-Armed Bandits, 2014. I would be surprised if one cannot apply SE/SAR style algorithms for the problem that the authors' are concerend?

I will adjust the score based on the feedback.

minor comments

* I would have one displayed equation or table or figure that compares LLR, full elimination, selective elimination like next to each other.
* In the end of section 3.1, “the reason for having different sets of active pieces for stopping and sampling is that it allows us to derive guarantees on expected stopping times. If one is only interested in high-probability results, it is possible to use the same set for both components.” ⇒ this is not immediately clear.
* Section 3.1's description can be improved. It would also be beneficial if we have a protocol/pseudocode description of what we expect from existing algorithms and what we do for eliminating the pieces, side by side.
  * Actually, it'd be better if there is a specific example algorithm where you show how elimination pieces at sampling is applied.
* It would be great if we know what stp and smp stands for in notations.
* definition of $\tilde{\mathcal{P}}\_t^{smp}$ : for the first line, did you mean to use $\alpha_{t,\delta}$ where $t$ is actually the most recent reset time j(t)? The current definition seems like it always computes $\overline{\mathcal P}\_t(i; \alpha\_{t,\delta})$ all the time anyways.
  * I also did not understand the purpose of this the reset times. why are we doing resets? is it because we will be discarding samples?
  * for reset times, are we using just the samples from the last block of time steps to compute $\overline{\mathcal P}$ or using entire samples ?
* Line 726, an union ⇒  a union
* Line 738, we the following ⇒ the following



**Strengths And Weaknesses:**

Strength is that the proposed strategy leads to improved empirical performance. The weakness is that the theory does not seem to be strong, at least in the current form.

The contribution is clear, but the details can be explained better. So, I would say that clarity is below expectation. Originality, quality, and significance is all mediocre (just around the bar).

My main complaint is that there is no 'provable' improvement of the proposed methods in mathematical analysis. I do not mean the proof for 'not worse', but a proof like 'a strictly upper bound can be achieved and it can make an orderwise difference in some cases'. Empirical evaluation shows that the empirical performance improvement can be better, but in my opinion, it is rather a minor improvement.

----
(after rebuttal) I have changed the score as the authors have addressed the concern.

---

> ### Author Response · Authors · 2022-08-01
> **Answer to Reviewer W482**
>
> We thank you for your time reviewing our paper and for your suggestions. There were two points raised by most reviewers, which we answered in a general comment: please read it first, then read this answer.
>
> **How $H^\star$ is related to $H_p$**
>
> We thank the reviewer for suggesting that we illustrate this on an example. In Appendix H of our revision, we added an example of a linear BAI problem where we can show that the bound (8) on the elimination times of certain arms is orderwise smaller than the bound (9) on the stopping time (see Proposition H.5 in Appendix H.3 and the remark immediately below it). The instance is a simple linear bandit problem with $d=2$ and arbitrary $K \ge 3$. The optimal complexity $H^\star(\theta)$ is fully characterized by the first two arms (the optimal one and an $\epsilon$-optimal one) and it scales roughly as $\epsilon^2$. On the other hand, for all other arms $k>2$, any $\omega$ close to the optimal proportions yields $H_k(\omega,\theta) \ge O(1)$. This implies that, from Theorem 2.6, all such arms can be eliminated with a complexity that can be made arbitrarily smaller than the stopping time $\log(1/\delta)/\epsilon^2$ as $\epsilon$ and $\delta$ go to zero.
>
> To better formalize how this reflects in provable gains in computation time for elimination-based strategies, in Appendix H.2 we analyze the behavior of an “oracle” sampling rule that plays the optimal allocation from the lower bound (which is also present in our experiments), a strategy for which the computation time is dominated by the stopping rule. As we show in Proposition H.3, in this instance both selective and full elimination can be made arbitrarily faster than LLR stopping as $\epsilon$ and $\delta$ decrease.
>
> We are happy to include both these examples in the final version of the main paper.
>
> > The explanation of reset times in Section 3.1 was not clear at all to me. What is the role of the resetting?
>
> We added a discussion about this at the end of Section 3.1 in our revision. We need the resets for two main reasons. First, they ensure that the algorithm stops almost surely as required to guarantee $\delta$-correctness (see Def. 1.1). Without resets, it might happen with some small (less than $\delta$) probability that pieces containing the true parameter are eliminated (eg, in BAI this means that the optimal arm is eliminated). In this case, the sampling rule might stop exploring relevant directions and never recover, hence preventing the algorithm from stopping. The second reason is more technical: resets guarantee that the thresholds $(\alpha_{s,\delta})_s$ used for computing ${\mathcal P}_t^{smp}(i)$ are within a constant factor of each other. This is crucial to relate the LLR of active pieces at different times, which in turn is crucial for proving Theorem 3.2.
>
> > please provide a specific example algorithm where you show how elimination pieces at sampling can be combined
>
> In Section 3.1 of our revision, we added an example of how track-and-stop can be combined with elimination at sampling and stopping (see Algorithm 1). We chose track-and-stop due to its generality (it handles general structures and queries) and since we believe its sampling rule to be well representative of all other approaches we consider, as most of them either implicitly or explicitly solve the optimization problem from the lower bound (i.e. they compute $H^\star(\theta)$ and/or $H_p(\omega,\theta)$ for different pieces). In Appendix I, we added another example for linear BAI, using LinGame as our base algorithm.
>
> > How existing elimination strategies are not easy to extend to combinatorial settings: I think Successive Accept and Reject (SAR) was applied to combinatorial pure exploration in Chen et al.
>
> We thank the reviewer for pointing out that algorithm (we mentioned it in the revised paper). Note that we are not claiming that designing elimination strategies for combinatorial settings is impossible, but rather that it is not easy at the level of generality we consider. The CSAR algorithm of Chen et al. works for a specific combinatorial setting in unstructured bandits, while we tackle possibly any pure exploration query in general structured (e.g., linear) bandits. On the other hand, adaptive algorithms like the one of [Non-asymptotic pure exploration by solving games, Degenne et al 2019] can be applied readily to all settings we consider.

---

> > ### Author Response · Authors · 2022-08-01
> > **Answer to minor comments of Reviewer W482**
> >
> > > In the end of section 3.1, “the reason for having different sets of active pieces for stopping and sampling [...]” ⇒ this is not immediately clear.
> >
> > When deriving guarantees on the expected stopping time, the stopping and sampling rules are analyzed under two different concentration events: Equation 3 for the former and Equation 6 for the latter. The corresponding elimination rules need to be defined with thresholds coherent with those used in such “good events” (as we need to guarantee that the true parameter is never eliminated whp), hence yielding two different sets. If we are only interested in high probability bounds on the stopping time, then both rules can be analyzed under the concentration event of Equation 3, and a single elimination rule based on its threshold can be defined. We will add this discussion to the paper if the reviewer thinks that it is more clear.
> >
> > > It would be great if we know what stp and smp stands for in notations
> >
> > “stp” stands for stopping and “smp” stands for sampling.
> >
> > **About the definition of the smp sets**
> >
> > The definition is correct, as
> >
> > $\tilde{P}_t^{\text{smp}}$
> >
> > is updated at every step by eliminating from
> >
> > $\tilde{P}_{t-1}^{\text{smp}}$
> >
> > using the current threshold $\alpha_{t,\delta}$, except for reset times, where eliminations restart from the full set of pieces.
> >
> > > for reset times, are we using just the samples from the last block of time steps to compute P― or using entire samples ?
> >
> > We can use all the samples, there is no need to discard them.

---

> > > ### Comment · Reviewer_W482 · 2022-08-08
> > > **thanks for addressing the issue**
> > >
> > > thank you for addressing the issue. I have raised the score accordingly.

---

### Author Response · Authors · 2022-08-01
**General Answer**

We thank all reviewers for their valuable feedback. We address two concerns which have been raised by most reviewers: (1) the clarity of our presentation of elimination at sampling (Section 3) and (2) the understandability of the main theorems; in particular, whether there are examples where the quantities in Theorem 2.6 can be computed explicitly, where the elimination times are provably much smaller than the stopping time, and where the computational complexity is provably reduced with elimination.

**Clarity of presentation**

To address the first concern, we uploaded a revised version of the paper in which the introduction of Section 3 and Section 3.1 are reworked (pages 6-7). The main changes are:

- as all reviewers suggested, we included the *pseudo-code of Track-and-Stop* with and without elimination, side by side for easy comparison;

- the presentation of the sampling sets is reformulated (this is also supported by a pseudo-code in Appendix I which we could not include in Section 3 due to space constraints, but would be included in a final version);

- the assumption on the shape of the threshold beta was pushed to appendix E to make room for the code.

**Understandability of the theorems**

To address the second concern, we added in Appendix H an example of a linear BAI instance where we can compute explicitly the quantities appearing in Theorem 2.6. For this specific instance, we prove that

- *Proposition H.4*: the computational complexity of a strategy playing the optimal proportions from the lower bound is provably reduced with elimination w.r.t. LLR stopping (and the former can be made arbitrarily faster than the latter as the problem complexity increases);

- *Proposition H.5*: any low information regret sampling rule (as in the context of Th. 2.6) provably discards most arms way before the stopping time.

We know that reading revised versions of papers is a lot of work, so we made sure that the changes were mostly localized to a small part of the main paper (Section 3, less than one page) and to a self-contained appendix (App. H). Apart from Section 3, the only change we made to the main paper is some information added to Table 1 to highlight the gains in computational complexity. All our modifications are highlighted in blue.

---

### Meta-Review · Area_Chair_4gcg · 2022-08-27

**Recommendation:** Accept
**Confidence:** Less certain

**Metareview:**

This paper has initially received borderline scores: the reviewers appreciated the general algorithmic framework and the high technical quality, but some of them lamented the relative weakness of the contribution (in particular the lack of hard improvements over existing results) and pointed out that the presentation could use some improvements. Some of these concerns were addressed in a revised version of the paper and a series of well-written author responses, which eventually convinced several reviewers to raise their scores.

Eventually, all reviewers agreed that the paper is acceptable for publication. The authors are encouraged to do another pass of revision when preparing the final version of the paper, and take all the reviewers' comments into account in the process.

**Award:**

No

---

### Decision · Program_Chairs · 2022-09-14

Accept